# Numerical Analysis of the Radial Load, Pressure and Velocity Fields of a Single Blade Pump

Dávid Bleho *, Róbert Olšiak , Branislav Knížat and Marek Mlkvik 

Faculty of Mechanical Engineering, Institute of Energy Machinery, Slovak University of Technology in Bratislava, Námestie Slobody 17, 812 31 Bratislava, Slovakia
* Correspondence: david.bleho@stuba.sk

**Abstract:** The centrifugal screw-type pump is a type of pump which, due to its hydraulic and mechanical properties, is used in several areas of the industry (e.g., for sludge and rainwater disposal). To avoid impeller passage clogging, the 3D impeller geometry is designed as a helically curved blade added to a conical hub. The passability through the fluid canal of the modelled impeller is 100 mm. In this paper, the magnitude of the radial force on an impeller blade is investigated as a function of the flow rate. The digital model was designed in Catia V5 and calculated using the commercial Ansys CFX software. A numerical computational fluid dynamics (CFD) method was used to investigate the performance characteristics of the pump, specifically discussing internal flow conditions such as velocity, pressure and the radial force mentioned above.

**Keywords:** single blade impeller; CFD; pressure field; velocity field; radial force



## 1. Introduction

The single blade pump is suitable for use in a wide range of applications. The advantage of this type of blade over conventional blades is the high throughput, which is measured by the passage of a spherical model passing freely through the flow channel. Therefore, fibres and solid particles up to an appropriate size can be transported without clogging.

During the operation of a pump, there is a complex three-dimensional unsteady flow inside. The greatest impact on the performance of the unit is hydraulic excitation. Static and dynamic interference between the impeller and the volute is the main reason for the radial force generated inside the centrifugal pump [1].

However, avoiding flow disturbances caused by clogging due to the geometry of the minimum number of blades introduces the obvious problem of hydrodynamic imbalance, which was studied by Aoki et al. [2]. Mainly because of the rotor–stator interaction, the flow around the blade produces a strongly asymmetrical unsteady fluid flow. This phenomenon leads to periodic hydrodynamic force acting on the impeller surface and results in strong vibrations of the rotor [3]. Predicting such a flow field with the required accuracy is hopeless using known analytical methods. In the past decade, numerical methods have been developed solving the Reynolds-averaged Navier–Stokes (RANS) equations. Considering the problem of centrifugal pump flow, the computational fluid dynamics (CFD) method is becoming more and more popular. With the available computer hardware, it is possible to implement consequential numerical solutions for multiple flow fields in a time- and cost-effective manner, even for unsteady flows. So far, considerable research and discussion has been carried out by combining the characteristics of the internal flow field, pressure, and vorticity obtained by different calculation methods compared to experimental results [4–6]. Despite this, details of the behaviour of the radial loads in a screw-type centrifugal pump are not yet clearly understood, because its magnitude depends on the properties of the non-uniform flow through the pump and, especially, at the near-tongue region of the volute.

## 2. Definition of Problem

An important step in the design of spiral hydrodynamic pumps is determination of the hydrodynamic radial force. The radial force in a hydrodynamic pump is determined by the nature of the flow in the impeller outlet and the diffuser inlet. The flow field downstream of the impeller is usually highly non-uniform and causes pressure pulsations as the stator blade is bypassed, resulting in dynamic forces. A special kind of stator is the spiral, which may be considered as a single blade diffuser. Here, asymmetrical pressure and velocity fields are generated at flows outside the BEP (best efficient point), which result in radial loading. The critical case is the single-bladed impeller, where the effects of the pressure forces do not cancel even in the BEP. The pressure and velocity fields behind such an impeller are strongly non-uniform and asymmetric, resulting in sustained radial loading in each flow rate. However, knowledge of these forces is critical for the design of the shaft and bearings, as well as for the life cycle of the entire pump. Empirical or semi-empirical equations of various authors (e.g., Stepanoff [7] or Biheller [8]), which are now considered classical in the literature, have been derived under certain general assumptions that may not be satisfied in particular cases. Another opportunity is to determine the force for a particular impeller shape. Figure 1 shows the silhouette of the radial impeller. The radial force acting on the impeller of this shape is a combination of the pressure action on its outer surface and the hydrodynamic impact of the flowing fluid due to a change in its momentum. Parts of the radial force $F_x$ and $F_y$ can be described by Equations (1) and (2), respectively.

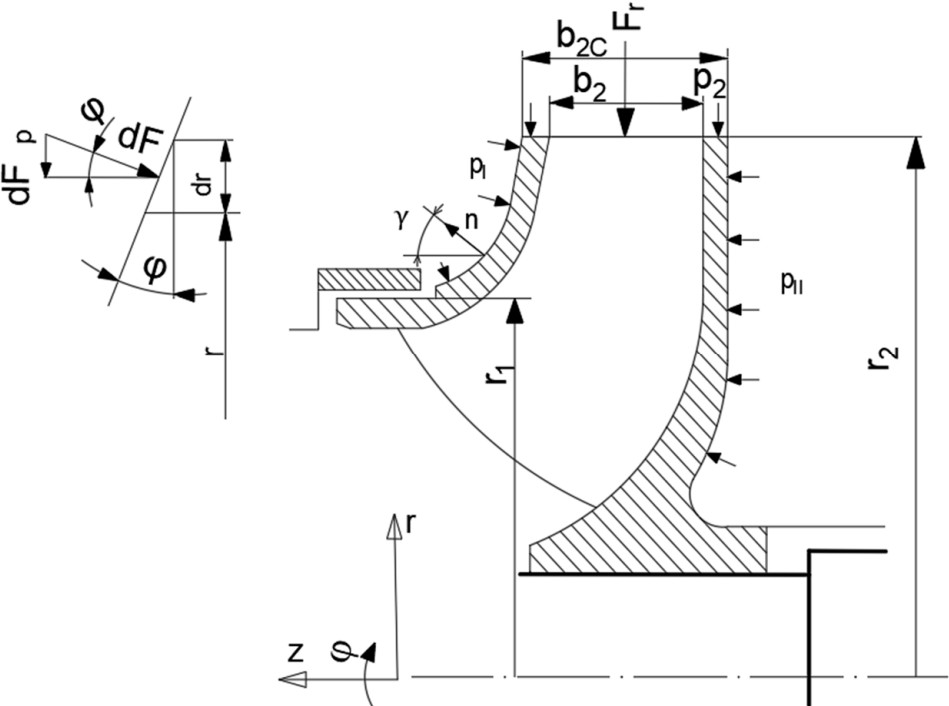

**Figure 1.** Pressures and force acting on rotor outside surfaces [9].

The components $x$ and $y$ of the radial force depend on the pressure distribution around the impeller.

$$F_x = -b_{2C} \, r_2 \int_0^{2\pi} p_2(\varphi) \, \cos(\varphi) \, d\varphi$$

$$- \rho \, b_2 \, r_2 \left( \int_0^{2\pi} c_{r2}^2(\varphi) \, \cos(\varphi) \, d\varphi - \int_0^{2\pi} c_{r2}(\varphi) c_{u2}(\varphi) \, \sin(\varphi) \, d\varphi \right)$$

$$- \int_0^{2\pi} \int_r^{r_2} \left[ p_2(\varphi) - \rho \, \frac{\omega^2}{8} \left( r_2^2 - r'^2 \right) \right] \mathrm{tg}\, \gamma(r) \, r \, dr \, \cos \varphi \, d\varphi \tag{1}$$

$$F_y = -b_{2C} \, r_2 \int_0^{2\pi} p_2(\varphi) \, \sin(\varphi) \, d\varphi$$

$$- \rho \, b_2 \, r_2 \left( \int_0^{2\pi} c_{r2}^2(\varphi) \, \sin(\varphi) \, d\varphi + \int_0^{2\pi} c_{r2}(\varphi) c_{u2}(\varphi) \, \cos(\varphi) \, d\varphi \right)$$

$$- \int_0^{2\pi} \int_r^{r_2} \left[ p_2(\varphi) - \rho \, \frac{\omega^2}{8} \left( r_2^2 - r^2 \right) \right] \mathrm{tg}\, \gamma(r) \, r \, dr \, \sin \varphi \, d\varphi \tag{2}$$

Here, the total radial force is calculated as follows: $F_r = \sqrt{F_{rx}^2 + F_{ry}^2}$. Equations (1) and (2) include parameters that express geometry ($b_{2C}$, $b_2$, $r_2$, $r'$). Obtaining these equations is not a problem, because the shape of the impeller is known. In addition, pressure and velocity fields appear later in these equations (parameters ($p_2(\varphi)$, $c_{r2}(\varphi)$, $c_{u2}(\varphi)$). These fields need to be specified, i.e., calculated or measured. In ref. [10], an efficient computational procedure is proposed for a one-dimensional spiral flow model for this purpose. However, this approach does not provide satisfactory results because the experiment and calculation do not match quantitatively. On the basis of a hypothesis, the idea will always be incomplete and will not sufficiently reflect the actual phenomena taking place in the pump. For this reason, CFD methods have become necessary to refine the calculation. Additionally, CFD computation is dependent on the choice and settings of the mesh, boundary conditions, turbulence model, etc.

The value of radial force is also influenced by the type of casing (single-volute, double-volute, diffuser) and whether the pump is operated at the BEP. Its values are the highest in single-volute pumps at operating points far from the BEP. A pump operating outside the tolerable flow range can suffer from bearing problems or even shaft collapsing [11].

Appreciably essential facts need to be considered here, in comparison with conventional centrifugal pumps. With regard to the asymmetric blade shape, the computed blade designed to accomplish energetic parameters causes dynamic unbalance of the whole rotational part of the pump.

The impeller under study is a single blade impeller with a balancing mass added to the rear disc for static and dynamic balancing. The removal of the mass from the front leading edge is related to the required passability that the impeller must meet. The main geometric data of the single blade pump under investigation and the operating conditions are summarized in Table 1, and the shape of the investigated impeller with its meridian section is shown in Figure 2.

**Table 1.** Specifications of the single blade centrifugal pump.

| | Parameters | Value |
|---|---|---|
| Impeller | Angle of the location of $R_{\max}$ | 342° |
| | Blade width | 10 mm |
| | Maximum radius $R_{\max}$ | 110.9 mm |
| | Total angle along the blade | 443° |
| | Passability | 100 mm |
| Suction pipe | Diameter $D_1$ | 108 mm |
| Discharge pipe | Diameter $D_2$ | 100 mm |
| Rotating condition | Flow rate (at BEP) | 32 l s$^{-1}$ |
| | Rotating speed | 1450 min$^{-1}$ |

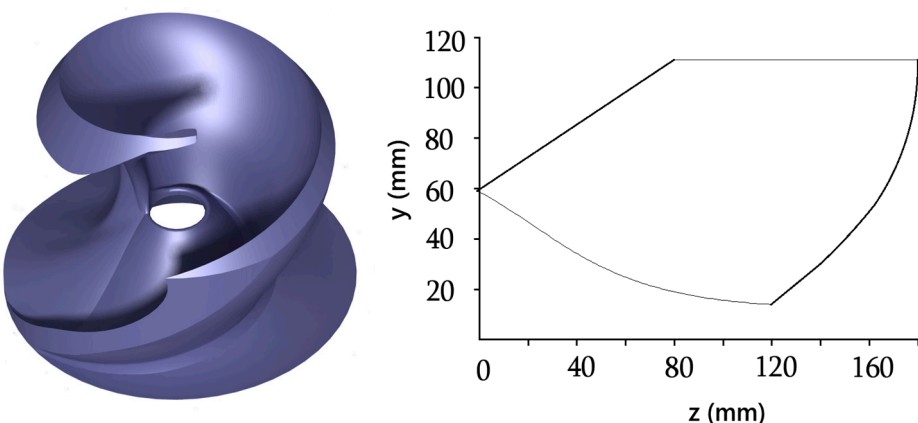

**Figure 2.** Investigated impeller and its meridian section.

## 3. Numerical Model

CFD simulation is a complex process and involves the following individual steps: the pump fluid field model was created using Catia V5, the Ansys Meshing module was used to mesh the geometrical model, and finally numerical calculations were performed using Ansys CFX software.

### 3.1. Meshing

Ansys Meshing is a meshing tool that allows a relatively simple generation of mesh, while achieving acceptable quality. The aim was to deploy a pump model with sufficient mesh quality, but at the same time not to exceed the total number of elements at the level of 8 mil, due to the length of the calculation. The mesh was generated as unstructured and consisted of tetrahedral elements (impeller and spiral volute), and hexahedral elements (suction and discharge pipes) (Figure 3). The density of the mesh was analysed in ref. [12]. This was a pump of the same type (single blade), and the authors also verified the CFD simulation by experiment, and the agreement between the two approaches was very good. Therefore, the results from the paper [12] have strong relevance for our pump case. We used twice the number of elements than the authors of the mentioned paper. The total number of grids in the stationary domain is about 3,381,000 (suction pipe—798,120, spiral volute with discharge pipe—2,583,682), and in the rotation domain is 3,830,556 (impeller). In the region close to the walls and in the gaps between the tip of the rotor blade and the pump body, the computational mesh was substantially densified. The size of the elements was in the range from 4 to 8 mm, with the "proximity" function enabled, which automatically thickens the mesh in places with complicated geometry and the "curvature" function, providing the correct structures of the mesh elements according to the curvature of the entity, so that the elements do not violate the curvature in that entity. The "sizing" function was used to densify the mesh in the blade edge area with the 0.25 mm element size. The boundary

layer was applied to all pump circulating surfaces with the thickness of the first layer set at 0.05 mm and the number of layers set at 15. The layer thickness had a value of 1.2.

Knowing the velocity profile close to the wall is very important in terms of correct meshing of the simulated domain. The surface quality corresponds to the situation near the wall zones. They were treated with scalable wall functions, which can be easily adjusted to various values of $y+$ and facilitate mesh refinement. The value of $y+$ is less than 40, which is sufficient for the RANS model applied here, because high pressure gradients are expected. The mesh generation and computational model are described in more detail by De Souza et al. [13].

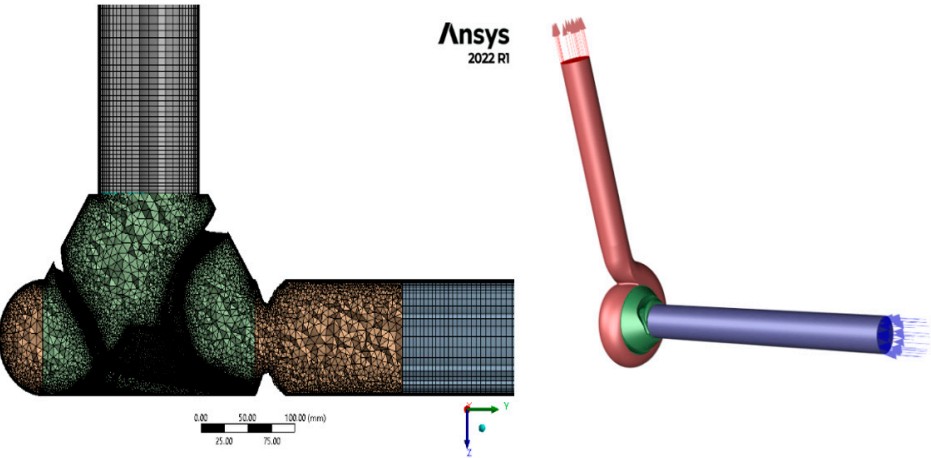

**Figure 3.** Mesh and modelled computational domains.

### 3.2. Turbulence Model

The BSL turbulence model was developed by combining two basic RANS models: the standard k-epsilon and the Wilcox k-omega model. These models reproduce the log-layer without near wall corrections and, thus, the EARSM approach may be invoked without the wall damping function. Obviously, this model does not reproduce the correct near wall behaviour for Reynolds stresses, but that may be less important in high Reynolds number flows, and this approach is an alternative that somewhat simplifies the formulation [14]. The simulated model was based on the solution of the RANS equations for the flow of an incompressible fluid, and the BSL EARSM turbulent model was implemented, which shows relatively accurate results with increased pressure gradient and flow with boundary layer detachment, making it suitable for simulations of, for example, turbomachinery.

### 3.3. Solver Setup

To properly define the boundary conditions, the suction and discharge pipes were also modelled. The part of the suction pipe must be long enough to avoid intense secondary phenomena on the inlet side of impeller [15]. The mass flow rate boundary condition was set at the inlet of the pipe, and the atmospheric pressure was set at the outlet. The CFD model of the pump consisted of three domains: A—inlet pipe, R—impeller, S—spiral. Domains A and S were set as stationary, and domain R was assigned the corresponding pump rotational speed. Water was selected as the working medium and the rotor speed was 1450 rpm. As this is a semi-open impeller, the front disc area is set as a "counter rotating wall" to simulate the effect of a stationary front disc.

As follows from the analysis of the problem, the majority of authors of scientific papers dealing with the topic of simulation of single blade pumps describe the character of the fluid in the inter blade channel as highly nonstationary, and therefore it is necessary to use transient simulation. Based on research in the literature on unsteady (transient) numerical simulations of single-blade centrifugal pumps [13,16,17], the transient simulation was calculated. The interfaces between the rotor and stator parts of the computational domain were of the "transient rotor stator type". This predicts the actual transient flow interaction

between the stator and the rotor. In this approach, the relative transient motion between the components on either side of the interface is simulated. Accounting for any interaction effects between components that are in relative motion with each other, the interface position is updated at each timestep as the relative position of the elements of mesh on either side of the interface changes. The disadvantage is the large computing resources required. The number of internal cycles per iteration was set in the range from 4 to 6.

The temporal independence study for the choice of timestep value was carried out by De Souza in ref. [18] through a process of refinement and coarsening of the timestep size. The timestep of the unsteady calculations $\Delta t$ was set to $6.591 \times 10^{-4}$ s, and a time independence study was performed for $\Delta t/2$, $\Delta t$ and $2\Delta t$. De Souza found that only a marginal (<1%) change in the average head values was recorded in the range of timestep selection. Based on these results, we chose a timestep size of $6.89655 \times 10^{-4}$ s. One timestep corresponded to a $6°$ rotation of the impeller, thus 60 transient results are contained to calculate one revolution of the impeller. The general grid interface (GGI) is used to interface the static and dynamic part during CFX preprocessing. For a wall function, a smooth wall condition was used. Six full impeller revolutions were required for the individual operating point to steady the flow, which represents 360 iterations. A larger number of iterations would mean a large timestep, and that is time consuming. The monitored parameters that were evaluated were torque, specific energy, hydraulic efficiency, and radial force.

## 4. Numerical Calculation of Pressure Force

The pressure forces can be simply determined using a numerical model. Pressure and force are related, which means that one of them can be calculated, if you know the other, using a physical equation: pressure = force/surface. The pressure and the normal component of the viscous stress tensor $\tau$ are integrated over the surface+ $\Omega$ of the single blade impeller. The integration is carried out discretely, using $p$ and $\tau$ for the impeller surface. Then, the total normal force acting on the surface [19]:

$$F = \int_{\Omega} (-np + n \cdot \tau) D\omega \tag{3}$$

where $n$ is the normal unit vector of the surface, pointing into the fluid. Then, the pressure obtained from the numerical method includes both dynamic and hydrostatic parts.

The calculated specific energy across the pump was obtained by averaging the differential pressure between the inlet and outlet across the two revolutions of the impeller.

$$Y = \frac{\Delta p}{\rho} \tag{4}$$

The shaft power was calculated as the sum of the normal and tangential moments on the impeller surface about the axis of rotation. The hydraulic efficiency was calculated according to Equation (5):

$$\eta = \frac{\rho QY}{M\omega} \tag{5}$$

## 5. Numerical Results

The CFD model was used to investigate the performance parameters, the radial force acting on the impeller surface, and the pressure and velocity fields at three positions of the impeller. Based on the obtained pump characteristics (Figure 4), the lowest value of radial force in the BEP ($32 \, \mathrm{l \, s^{-1}}$) region can be observed. According to the radial force values, a curve and a quadratic equation describing the prediction of its further development were numerically predicted. The relatively high value of the radial force can be attributed to the asymmetric distribution of the pump medium in the interblade area.

Figure 5 shows the time course of the calculated instantaneous radial force amplitudes $F_{\mathrm{rad}}$ during six rotations of the impeller $n_{\mathrm{imp}}$, with the resulting root mean square (RMS) curve corresponding to the given waveform. The value of the energy carried by the signal

representing the radial force can be expressed in terms of the true RMS value per revolution of the impeller according to Equation (6):

$$F_{\text{rad,RMS}} = \sqrt{\frac{1}{N} \sum_{i=1}^{N} F_{\text{rad,i}}{}^2} \tag{6}$$

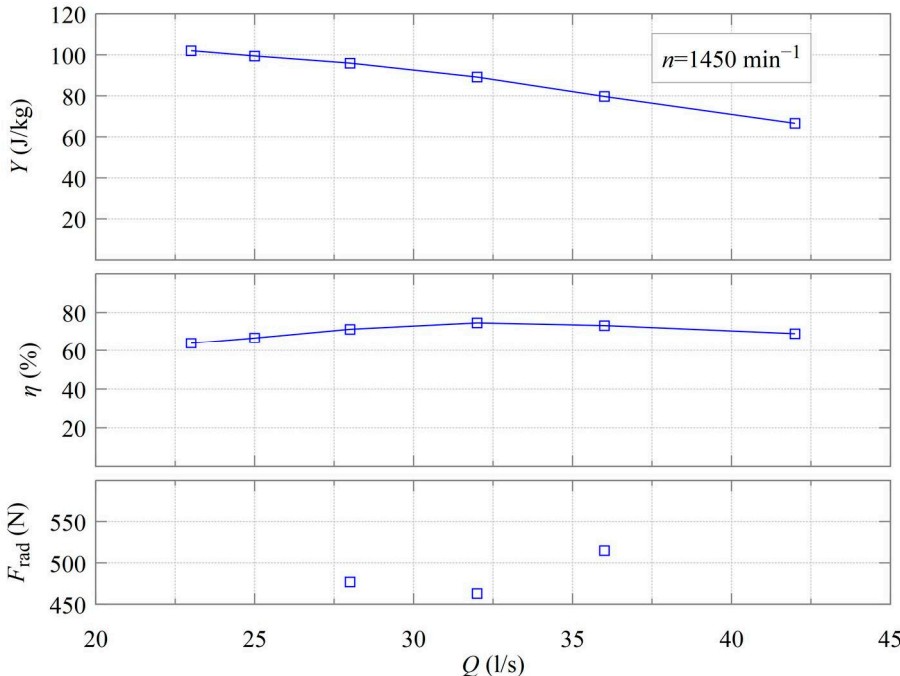

**Figure 4.** Pump performance curves and radial force waveform (calculate and prediction).

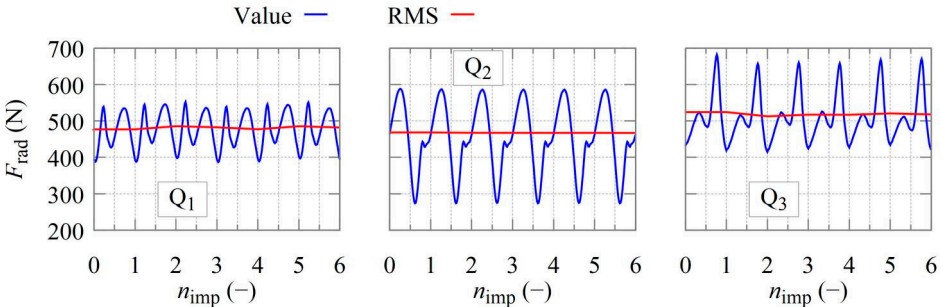

**Figure 5.** Radial force fluctuation during six revolutions of the impeller for flow rates $Q_1$, $Q_2$ and $Q_3$.

The simulation timestep was set so that one revolution of the impeller was divided into 60 rotations, corresponding to the number of instantaneous radial force amplitudes $N = 60$. For a particular flow rate, the simulation was run for multiple impeller rotations, so that the resulting radial force was determined as an estimate of the mean of the effective force values calculated from the effective force values of all the impeller rotations for a particular flow rate:

$$\hat{F}_{\text{rad, RMS}} = \frac{1}{M} \sum_{j=1}^{M} F_{\text{rad,RMS,j}} \tag{7}$$

Assuming that the simulation is performed in a steady state, the number of impeller rotations for the simulations was set to $M = 6$.

Each of the three curves represents the radial force waveform for a given flow rate (Table 2).

**Table 2.** Analysed flow rates and effective force values according to Formula (7).

| Flow Rates | Capacity | $\hat{F}_{\text{rad, RMS}}$ |
|:---:|:---:|:---:|
| $Q_1$ | $28\,\mathrm{l\,s^{-1}}$ | 481.577 N |
| $Q_2$ | $32\,\mathrm{l\,s^{-1}}$ | 467.408 N |
| $Q_3$ | $36\,\mathrm{l\,s^{-1}}$ | 518.242 N |

The periodicity of the waveform can be deduced, the phenomenon based on rotor-stator interactions between the impeller and the volute.

A significant peak can be observed for each oscillation with only one impeller blade. Only for a flow rate equal to $28\,\mathrm{l\,s^{-1}}$, two equivalent peaks are shown, with the smallest range compared to other flow rates. The radial force course at a nominal flow rate of $32\,\mathrm{l\,s^{-1}}$ oscillates to large values, but the resulting effective force value shows the smallest value.

The trend of the periodically oscillating hydrodynamic load can be observed in Figures 6–8, which show the contours of the instantaneous static pressure for three test flow rates 28, 32 and $36\,\mathrm{l\,s^{-1}}$.

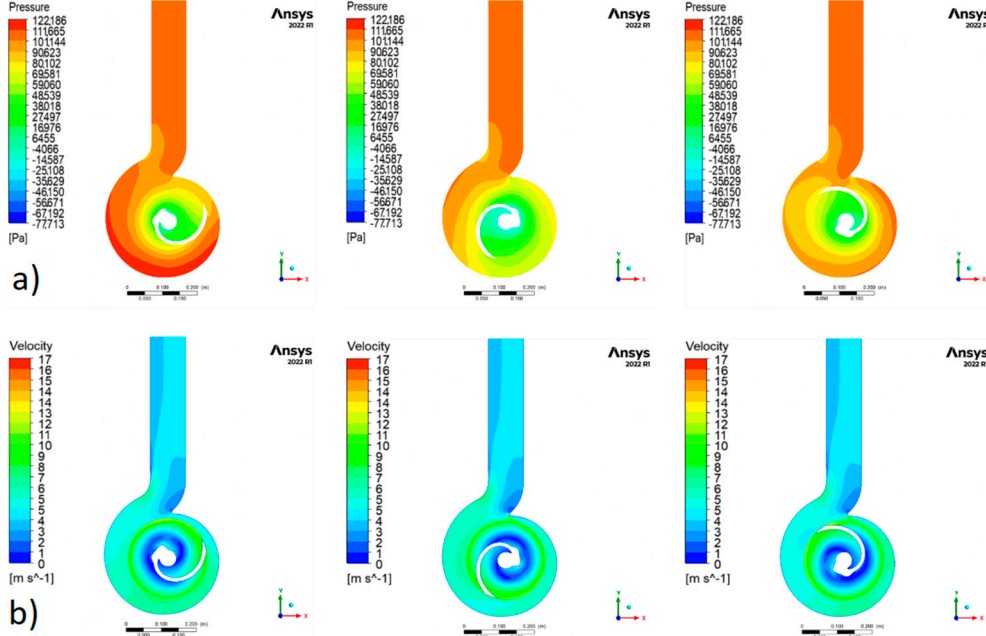

**Figure 6.** Pressure (**a**) and velocity (**b**) field at $28\,\mathrm{l\,s^{-1}}$.

The effect of the position of the impeller relative to the position of the spiral is shown in the figures above. The asymmetrically distributed pressure field shown in the plane perpendicular to the axis of rotation can be seen in Figures 6a, 7a and 8a. The position of the single blade impeller relative to the volute affects the pressure field, which rotates with the position of the impeller. The difference in impeller rotation between each frame is 120°. By observing them and then comparing them, one can see a clear characteristic of the flow nonstationarity. The figures were taken during the six revolutions of the impeller, when the flow was sufficiently steady.

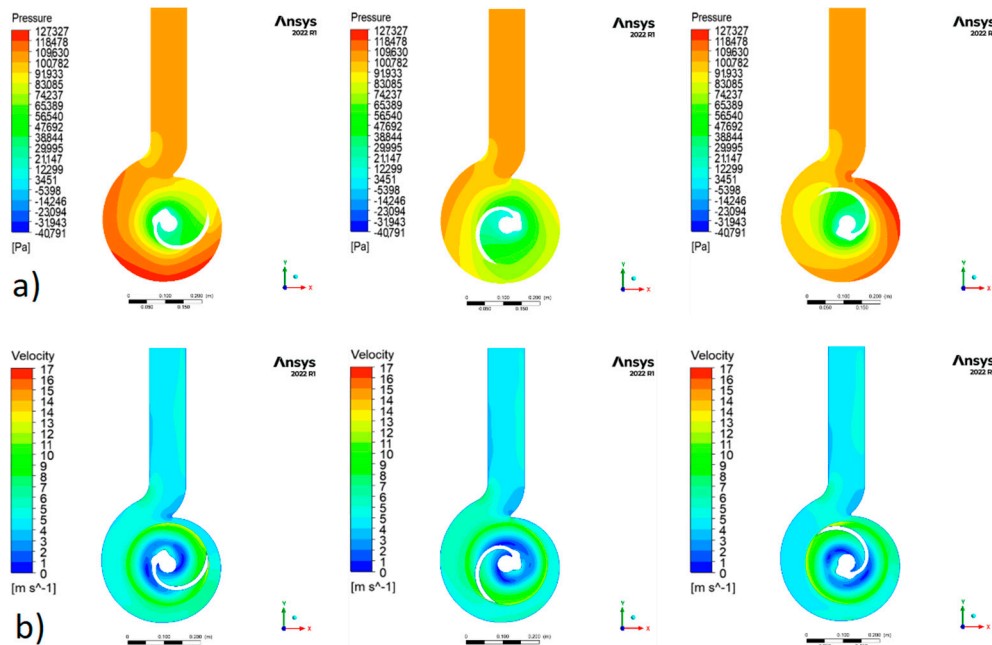

**Figure 7.** Pressure (**a**) and velocity (**b**) field at $32 \, \text{l s}^{-1}$.

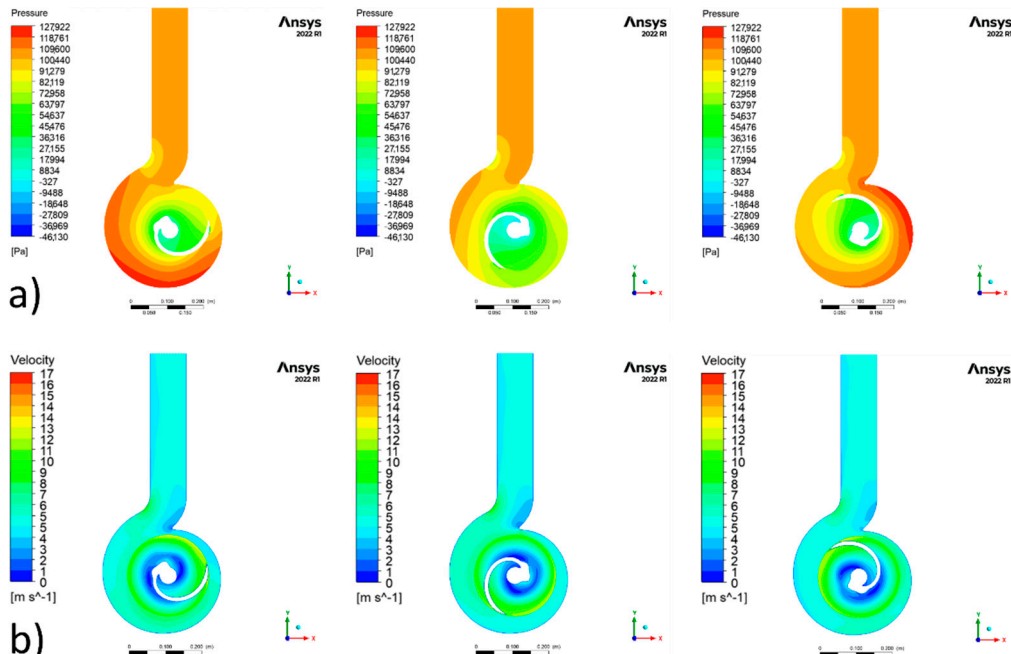

**Figure 8.** Pressure (**a**) and velocity (**b**) field at $36 \, \text{l s}^{-1}$.

Due to the low slip factor [20], the difference in bypass velocity between the suction and pressure edges of the blade is visible (Figures 6b, 7b and 8b). The region of darker blue colour, above the nose of the volute, represents slower flow, or this is where the swirling of the flowing fluid occurs as it bypasses the nose of the volute.

A steady velocity profile can be observed at the outlet of the volute neck, indicating low current pulsations during pump operation. It is also possible to observe the difference in bypass velocity between the suction and discharge sides of the blade.

## 6. Discussion and Conclusions

In our research, we focused on the CFD analysis of radial forces in a single blade pump with a spiral casing. We performed the calculation as a transient simulation at three flow rates around the BEP. The basis for the accurate determination of the instantaneous value of the radial force lies in the determination of the pressure and velocity fields, because according to relations (1) and (2), the components of the radial force depend mainly on them. Therefore, for the observed pump operating regimes, we present these fields in the Figures 6–8.

Workspaces were designed and analyzed by the CFD method with the CFX module. Time-independent simulation was used because of the significant interaction between the rotor and stator positions. The position of the impeller is closely related to asymmetrically distributed pressure and velocity fields. This phenomenon results in increased vibration and thus increased noise of the machine. For maximum machine life, the pump should be operated at nominal flow because the magnitude of the radial force is lowest here. According to the known progression, it is assumed that the growth of the radial force increases from the BEP point.

Higher static pressure represents a stronger pumping capacity, which is more appropriate for the delivery of nonhomogeneous and viscous media due to its self-non-clogging capability. On the contrary, higher pressure makes the transported material more susceptible to damage.

The low slip factor [20] is the difference in bypass velocity between the suction and pressure edges of one blade. The low-velocity region is around the nose of the spiral, where the fluid is probably swirling. At the outlet of the discharge neck of the volute, the overflow velocity gradually stabilizes so that the flow from the pump does not pulsate.

## 7. Suggestions

This paper reveals the power characteristics and internal flow field of a centrifugal single-blade centrifugal pump with operating conditions and their surroundings through numerical calculations. However, while research work has achieved certain results, there are still many problems that can be studied and need to be further expanded and deepened in future work. In the future, we will start working on an experimental analysis, and the research results of this paper can be compared with experimental ones. Another subject of investigation should be the known direction of the resultant radial force vector, the assessment of the hydraulic loads in the frequency domain, or the design of the mechanical/hydraulic balancing of the radial force for a single blade pump.

**Author Contributions:** Conceptualization, D.B., R.O. and B.K.; methodology, R.O. and B.K.; writing—original draft preparation, D.B. and B.K.; writing—review and editing, R.O. and M.M.; supervision, R.O. and M.M. All authors have read and agreed to the published version of the manuscript.

**Funding:** This work was supported by the Slovak grant agency KEGA, project No. 016STU-4/2022.

**Data Availability Statement:** The data presented in this study are available on request from the corresponding author. The data are not publicly available due to continuing research.

**Conflicts of Interest:** The authors declare no conflict of interest.

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
