# Peer review of "Numerical Analysis of the Radial Load, Pressure and Velocity Fields of a Single Blade Pump"

_2673-3951, doi:10.3390/modelling3040028_

Round 1

Reviewer 1 Report

The manuscript presents a CFD simulation of a single blade pump. The radial load, fluid pressure, and velocity fields can be obtained by the CFD simulation. However, the discussion on the results part is not enough for the readers to get the idea. I suggest the authors should make revisions to the numerical results section and address my following concerns:

1. In lines 115 and 165, the authors use ÷ to denote the ranges of parameters. Is it a mistake? Please change it.

2. In line 192, the authors state that the BEP is 32 litres per second. But in the x-axis of Figure 4, the unit is cubic meters per second. There must be a mistake here. Please make a change.

3. The discussion on Figure 4 is not enough. I suppose that the red line is the fitted radial force value, but the authors do not state it. Meanwhile, it lacks information on how to calculate the power and the efficiency. Please provide a detailed discussion on how the flow rate affects the four parameters. 

4. Also in Figure 4, the head and power are using the same axis, while the efficiency and force are using the same axis. It may cause some confusion. Maybe using two figures to show the trend of parameters is a good idea.

5. Figure 5 shows the force with different flow rates. I guess the dash lines are the RMS. But the RMS may not need to be shown in the figure. A simple table could be a better choice. 

6. In Figures 6, 7, and 8, the color bars for the pressure and velocity are lacking. Readers can not tell the values for the parameters. Please add the color bars.

There are still some grammar mistakes in the manuscript. Please check the paper carefully.

Reviewer 2 Report

The manuscript "Numerical Analysis of the Radial Load, Pressure and Velocity Fields of a Single Blade Pump" evaluates numerically the performance of a pump. The objectives of the investigation should be rewritten to clarify the purpose of each parameter evaluated. There isn’t a reported of the grid independence study, so there is not proof that the simulation was carried out takin in count this important point on numerical analysis. On the other hand, there is a mention that “the reduction of the timestep typically does not provide a better result”, but this assumption is taken right when it is supported by a temporal independence study, the one is missing in the present analysis. The results should be extended to understand the importance of each parameter evaluated.

Page 5 line 177: The Figure 3 isn´t mentioned on the text.

Page 5 line 180 – 185: The information was taken word by word from the reference 18.

Page 6 line 186 – 187: The information was taken word by word from the reference 18.

Page 7 line 212: The 26 should be change for 36.

Round 2

Reviewer 1 Report

All of my concerns have been addressed. The manuscript is ready to be published.

Author Response

We would like to thank the reviewer for his time and valuable comments.

Reviewer 2 Report

The link between the objective of the investigation and each parameter evaluated is not clear.

There isn’t a reported of the grid independence study.

There isn’t a reported of the temporal independence study, the authors explain the topic but not the reason for the value chosen.
